# The role of disulfide bonds in a *Solanum tuberosum* saposin-like protein investigated using molecular dynamics

**John H. Dupuis**[1], **Shenlin Wang**[2], **Chen Song**[3], **Rickey Y. Yada**[1]*

**1** Food, Nutrition, and Health Program, Faculty of Land and Food Systems, University of British Columbia, Vancouver, British Columbia, Canada, **2** College of Chemistry and Molecular Engineering and Beijing NMR Center, Peking University, Beijing, People's Republic of China, **3** Center for Quantitative Biology, Academy for Advanced Interdisciplinary Studies, Peking University, Beijing, People's Republic of China

* r.yada@ubc.ca

**Data Availability Statement:** Simulation trajectories (.trr, without water and ions), initial structures (.gro, .pdb), topologies (.top, .itp), and Gromacs molecular dynamics parameter (.mdp) files have been made freely available under the following DOI: 10.20383/101.0237; accessible at

## Abstract

The *Solanum tuberosum* plant specific insert (*St*PSI) has a defensive role in potato plants, with the requirements of acidic pH and anionic lipids. The *St*PSI contains a set of three highly conserved disulfide bonds that bridge the protein's helical domains. Removal of these bonds leads to enhanced membrane interactions. This work examined the effects of their sequential removal, both individually and in combination, using all-atom molecular dynamics to elucidate the role of disulfide linkages in maintaining overall protein tertiary structure. The tertiary structure was found to remain stable at both acidic (active) and neutral (inactive) pH despite the removal of disulfide linkages. The findings include how the dimer structure is stabilized and the impact on secondary structure on a residue-basis as a function of disulfide bond removal. The *St*PSI possesses an extensive network of inter-monomer hydrophobic interactions and intra-monomer hydrogen bonds, which is likely the key to the stability of the *St*PSI by stabilizing local secondary structure and the tertiary saposin-fold, leading to a robust association between monomers, regardless of the disulfide bond state. Removal of disulfide bonds did not significantly impact secondary structure, nor lead to quaternary structural changes. Instead, disulfide bond removal induces regions of amino acids with relatively higher or lower variation in secondary structure, relative to when all the disulfide bonds are intact. Although disulfide bonds are not required to preserve overall secondary structure, they may have an important role in maintaining a less plastic structure within plant cells in order to regulate membrane affinity or targeting.

## Introduction

Instead of possessing an active immune system, plants rely on a complex system of antimicrobial compounds to defend themselves from pathogens, many of which are protein-derived. One such class of compounds are aspartic proteases (APs). In plants, many, but not all, APs contain an additional primarily helical segment called the plant specific insert (PSI), which

https://doi.org/10.20383/101.0237. Data is hosted at the Federated Research Data Repository (FRDR, https://www.frdr-dfdr.ca), courtesy of the Canadian Association of Research Libraries (CARL) and Compute Canada.

**Funding:** RYY is supported by the Natural Sciences and Engineering Research Council (NSERC, https://www.nserc-crsng.gc.ca/) of Canada, grant number RGPIN-2018-04598. JHD was supported by a NSERC PGS D scholarship. The funders had no role in study design, data collection and analysis, decision to publish, or preparation of the manuscript.

**Competing interests:** The authors have declared that no competing interests exist.

possesses an antimicrobial action independent from the parent AP. The PSIs fall into a larger family of saposin-like proteins (SAPLIPs) which includes the human saposins, NK-lysin, granulysin, human lung surfactant protein B, and protozoon pores (*Naegleria*, *Entamoeba*) [1]. Note–PSIs are domain swapped with respect to other SAPLIPs (*i.e.*, sequentially, the N-terminal helices of the *St*PSI align to the C-terminal helices in other SAPLIPs, such as the saposins); this concept is demonstrated visually in S1 Fig [2]. Both PSIs and SAPLIPs share a common structural fold of four to five amphipathic helices and a set of conserved cysteine residues that form three disulfide bonds (one exception being granulysin, which has two). Collectively, these proteins all possess the ability to interact with membranes [1]. Most often, this action is restricted to acidic pH and anionic phospholipid-containing membranes [1, 3]. In PSIs, there is one disulfide bond connecting N- and C-termini, and two disulfide bonds between helices 2 and 3, immediately before a large unstructured loop (~25 amino acids) (Fig 1).

Despite the highly conserved nature of the disulfide bonds, the protein maintains an ability to interact with membranes, even with increased rigor, after the reduction of disulfide bonds using dithiothreitol (DTT) which was observed for several PSIs, including those in potato (*Solanum tuberosum*, *St*PSI), barley (*Hordeum vulgare*), cardoon (*Cynara cardunculus*), and *Arabidopsis thaliana* [5]. Earlier works have been carried out to study the relationship between the PSI's vesicle fusion activity and the presence of disuldxfide bonds [5]. Using a common experimental platform (1:1 16:0–18:1 phosphatidylethanolamine/16:0–18:1 phosphatidylserine vesicles at pH 4.5), PSI-induced vesicle size increases were monitored as a function of disulfide-state [5]. For each of the four aforementioned PSIs, the ability of the PSI to increase vesicle size improved after reducing the disulfide bonds, with respect to those with intact disulfide bonds. These experimental results were also similar to those seen with saposin C [6]. As observed using circular dichroism spectroscopy, reducing the disulfide bonds had minimal effects on the secondary structure of the *St*PSI with only minor changes in helicity reported after reduction with up to 5 mM DTT at room temperature [4]. Upon heating (95˚C, 5 min) and levels of DTT > 2.5 mM, a more substantial loss of helicity was observed. In saposin B, the removal of disulfide bonds resulted in an increased susceptibility to trypsin digestion [7]. The above results indicate that the disulfide bonds in SAPLIPs may play a role in preserving structure under extreme environmental conditions (high temperatures) or when acted on by outside forces (digestive enzymes), but simultaneously, under ambient conditions, may provide flexibility to the structure, allowing for enhanced membrane binding. Thus, it would be of interest to identify the structural factors responsible for the stability of a SAPLIP with disulfide bonds removed. Here, the *St*PSI will be used as a model SAPLIP.

In order to observe the minute details of the effects of disulfide bond removal on the structure of a protein, and determine the potential structural rationale for enhanced membrane interactions, a high-resolution technique is needed. Molecular dynamics (MD) presents itself as a well-established tool to realize this goal. As computational resources become more accessible, the use of MD simulations has increased in order to study delicate systems at atomic resolution. Previously, MD has been used to study the *St*PSI in monomeric and dimeric states in solution [8], and also in the context of membrane binding as a function of pH and membrane types [9]. Both saposin C [10, 11] and surfactant protein B [12–14] have been the subject of extensive MD studies to probe lipid-binding dynamics. Other SAPLIPs have also been probed using various MD methods, including saposin A to study gas-phase picodisc properties [15], saposin B to study inter-subunit packing [16], and the role that saposin C plays in the stabilization and activation of its associated enzyme, glucocerebrosidase [17].

In the present work, MD was used to study the effects of the disulfide bonds on the structure and dynamics of the *St*PSI in an open dimeric form (Fig 1A), the structure of which is similar to that of the experimentally determined structure of saposin C [18]. All analyses were

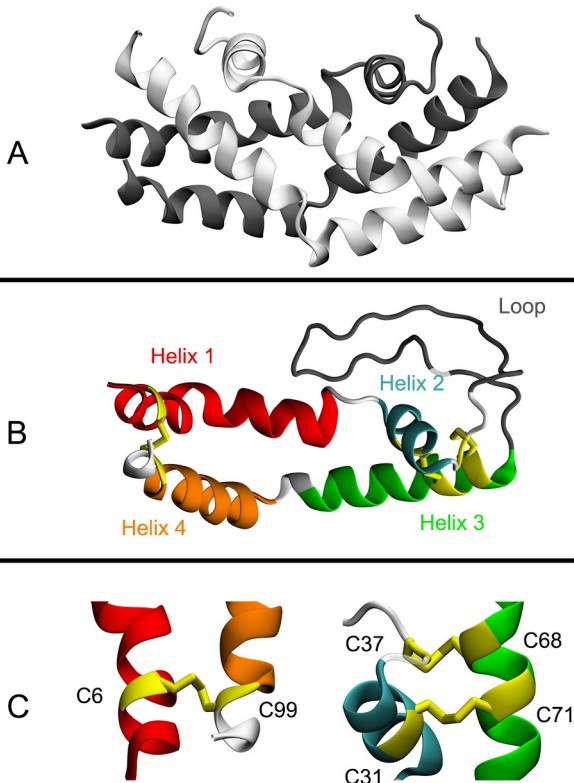

**Fig 1. Structural units of the StPSI.** A) StPSI in the "open dimer" conformation; separate monomers are different colors, and the loop portion has been omitted for clarity. B) The four helices and loop portion of a StPSI monomeric unit; each feature is color coded and all disulfide bonds are highlighted. C) Zoomed in view of the three disulfide bonds; helical elements are colored as per panel B. The initial structure for these renders is from the crystal structure of the StPSI (PDB ID: 3RFI) [4].

performed on a dimeric structure of the StPSI, however, where appropriate, data is reported for each monomer separately, an average of the two monomeric units, or for the dimer as a whole. One, two, or three disulfide bonds were removed in all possible combinations in order to ascertain the resultant effects on the structural stability of the StPSI. Root-mean-square fluctuation (RMSF), root-mean-square deviation (RMSD), and the radius of gyration ($R_g$) indicated that the StPSI is readily stable in solution, regardless of disulfide bond state. Stability was likely derived from inter-monomer hydrophobic interactions, and intra-monomer hydrogen bonds and salt bridges. It is postulated that removal of disulfide bonds instead alters the location and spread of variability in the local secondary structure which may have implications in membrane binding and *in vivo* targeting of plant aspartic proteases with the PSI segment still attached.

## Results

### Inter-sulfur distances

If two cysteine thiols are in close enough proximity, there is the potential to reform a disulfide. This possibility was explored by analyzing the distance separating Sγ bond partners on each monomeric unit of the dimer (Fig 1B). At pH 3.0, removal of the Cys6-Cys99 bond (Fig 2, blue curves), either alone or with one of the other disulfide bonds, results in a Sγ separation of approximately 5.0 Å (Fig 2, panels D1, D4, and D6). Removal of the Cys31-Cys71 (Fig 2,

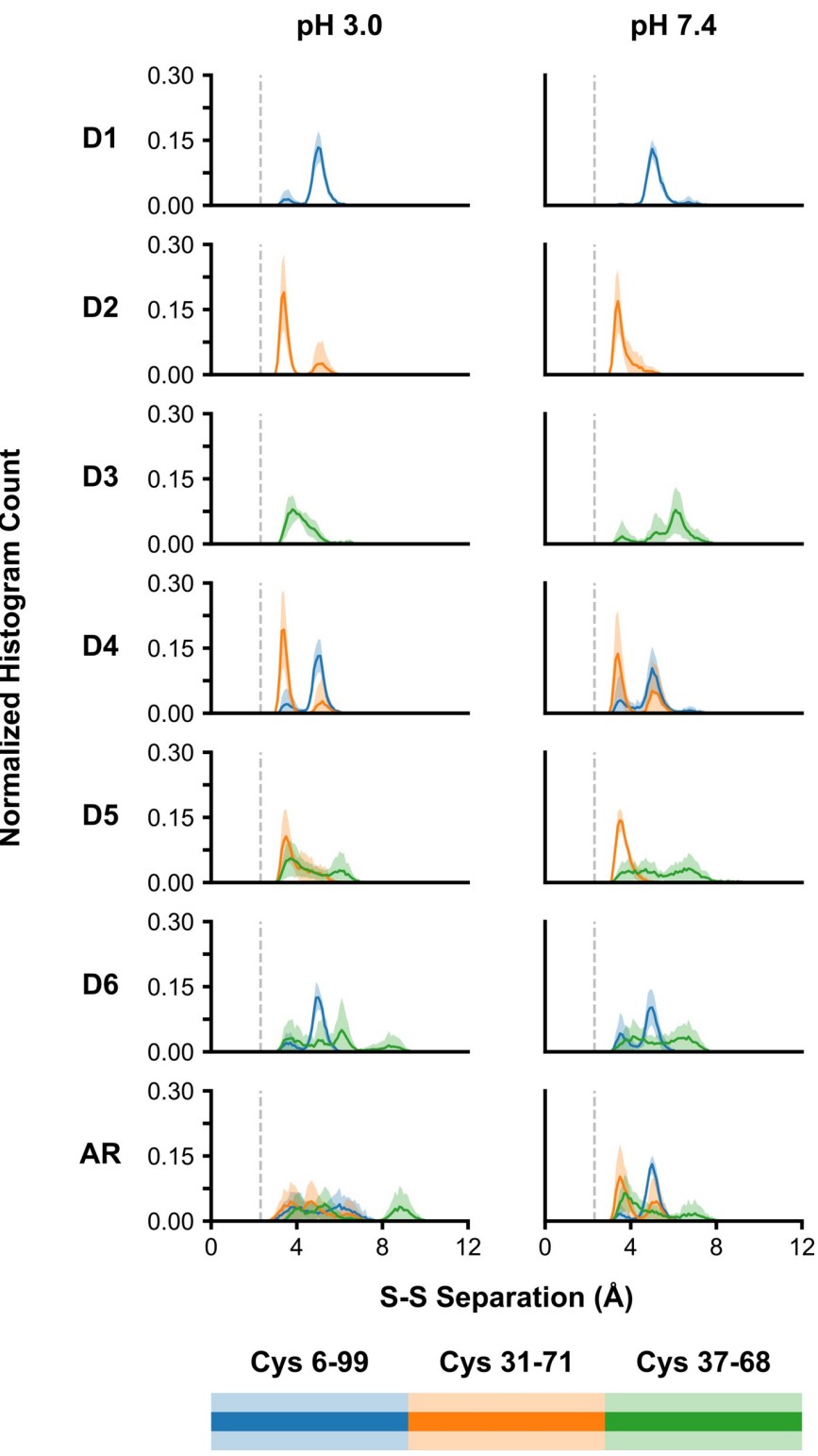

**Fig 2. Separation between native cysteine sulfur atom pairs after disulfide reduction at pH 3.0 and pH 7.4.** The native cysteine sulfur atom pairs were Cys6-Cys99 (blue), Cys31-Cys71 (orange), and Cys37-68 (green). Data were presented as a histogram over the last 10% of the simulations for each monomer and replicate. Dark lines are the average per-bin across the replicates and monomeric units (*n* = 6). Lighter shaded regions correspond to the per-bin standard deviation. The grey dashed line indicates the S-S distance in an intact disulfide bond (0.2038 nm).

orange curves) bond is likewise stable, but at a lower separation–hovering at approximately 3.5 Å (Fig 2, panels D2, D4, D5). Whether singularly or together, both Cys6-Cys99 and Cys31-Cys71 Sγ atoms maintain a stable separation once the disulfide bond is removed. The distance between Cys37-Cys68 Sγ atoms has proven to be much more sensitive to disulfide removal (Fig 2, panels D3, D5, D6, and AR). When removed alone, the separation appears to be pH-dependent. A pH 3.0, the separation between Cys37-Cys68 Sγ atoms is centered on ~4 Å, whereas at pH 7.4 separation is closer to 5 Å. Structurally Glu72 is in close proximity to this pair, which may explain the pH-dependent sulfur separation. At pH 3.0, this glutamic acid is neutralized, whereas at pH 7.4 it bears a negative charge, possibly leading to repulsion of one of the Sγ atoms in the pair. When Cys37-Cys68 is removed alongside Cys6-Cys99 (Fig 2, panel D6), much larger changes are observed. While the Cys6-Cys99 separations are similar to that of the D1 panel, the separation distribution of Cys37-C68 ranges from ~3 Å to 9 Å. Lastly, in the AR simulations with all disulfides reduced, dramatic changes were observed in the separation between Sγ pairs. At pH 3.0, the separation between both Cys6-Cys99 and Cys31-Cys71 Sγ atoms is spread over approximately 3–7.5 Å, and at pH 7.4, over a slightly smaller range of 3–6 Å. Lastly, and keeping in line with its relative instability, the Cys37-Cys68 Sγ separation ranges between 3–10 Å for both pHs Based on these results, using a separation of 4 Å as a cut-off, it is likely that the second disulfide bond, even if reduced, would likely reform into a disulfide. The possibility of the formation of non-native disulfide bond arrangements, or for disulfide interchange, was explored (S2 Fig). Based on the separation distributions of Sγ pairs, the possibility of forming a disulfide bond between Cys31 and Cys37, either after reduction or through disulfide interchange, cannot be entirely discounted. Conversely, the formation of a disulfide bond between Cys68 and Cys71 seems improbable.

## Protein structural stability

The *St*PSI in solution possesses extraordinary stability, both as a function of pH, and disulfide removal [4]. To probe the structural effects of removing individual disulfide bonds, normal mode analysis was performed over the last 10 ns of the simulations. The trajectory was fit to the first eigenvector, and the RMSF was calculated for each monomeric unit over the last 50 ns of the 100 ns simulations, and averaged together (Fig 3). The majority of the dimer's RMSF is localized to the loop portion. Unexpectedly, removing individual disulfide bonds did not perturb the RMSF profile substantially at pH 3.0 or pH 7.4. When the first disulfide bond is removed, there is an increase in RMSF in the C-terminus for pH 3.0. This is to be expected as Cys6 and Cys99 are both six or fewer residues from the termini. There are minimal non-loop RMSF changes when the second disulfide bond (Cys31-Cys71) was removed. Removal of the Cys37-Cys68 disulfide bond (D3), however, induced structure-wide destabilization, as evidenced by an increase in the standard deviation of the RMSF across nearly all residues. Interestingly, the RMSF and standard deviation for both of the free cysteines in this pair are quite low, compared to the rest of the structure. Removal of two disulfide bonds in combination did not have any appreciable impact on the RMSF profiles outside of the loop portion. In the AR simulations, the standard deviation profile was similar to what was seen in the D3 results, indicating that there are some residue-level perturbations occurring, albeit minor.

In order to gage potential global structural deviations, RMSD was calculated using the dimer backbone with and without the loop portion (Fig 4A). In both cases, only non-loop regions were used for the fitting procedure. The RMSD without the loop portion was stable at both pH over the last 10 ns of the simulations. At pH 3.0, the RMSD ranged from 0.183 ± 0.010 nm to 0.265 ± 0.026 nm, and at pH 7.4, from 0.201 ± 0.016 nm to 0.238 ± 0.016 nm (note the error values are the standard error of the mean, SEM). Interestingly, the maxima

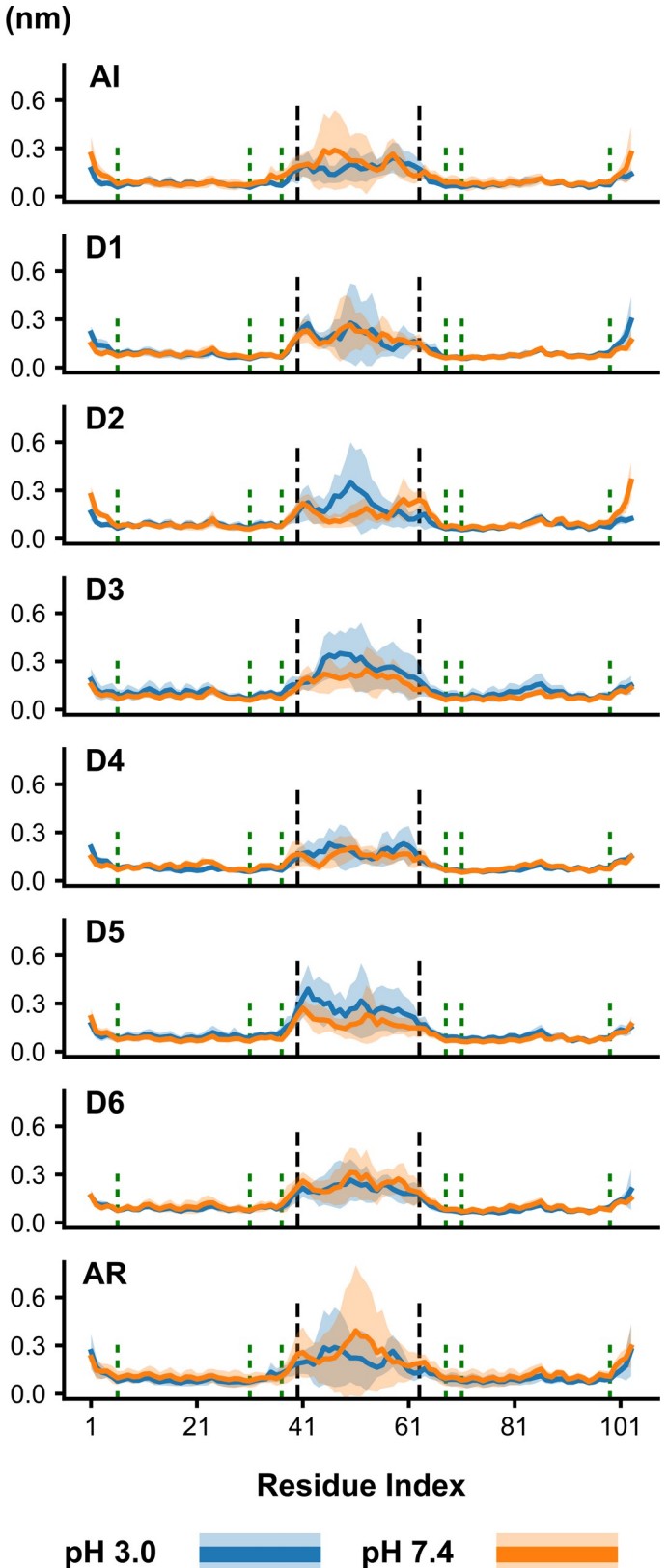

**Fig 3. Root-mean-square fluctuation over the last 10 ns of simulations.** Colored lines represent pH 3.0 (blue) and pH 7.4 (orange). Data are presented as combined average of both monomeric units and the three replicates ± standard

deviation (light shaded regions). Short vertical dashed green lines indicate cysteine locations; tall vertical dashed blank lines indicate the start and end points of the loop (residues 40–63 inclusive).

and minima are for the AI and AR simulations (RMSD without the loop), respectively, but within simulations D1 to D6, there is no trend of gradually increasing RMSD (Fig 4A). The RMSD of the full-length dimer varied more widely–from 0.472 ± 0.090 (pH 3.0, D3) to 0.816 ± 0.044 (pH 3.0, D6), indicating the stochastic nature of the loop. Further evidence to support the stability of the dimer, one independent 1 μs simulation for the AR configuration at both pH 3.0 and 7.4 was performed. Both of these simulations yielded RMSD values similar to those seen in the 100 ns simulations (S3 Fig). Similarly, the $R_g$ (Fig 4B) was calculated with and without the loop portion. In contrast to the RMSD, the $R_g$ for the dimer, with and without the loop portion, only increased modestly. At pH 3.0, the $R_g$ of the AI simulation, with and without the loop, were 1.733 ± 0.028 nm and 1.651 ± 0.007 nm (the error being SEM), respectively. The consistency between the full-length analysis and the analysis without the loop was a trend carried across all simulations at both pH values, with the maximal difference between the two measures as 0.19 nm.

The secondary structure variation was calculated for all simulations (See Methods, Section 4.3.4. for details of the calculation), with the results for only AI and AR simulations at both pHs being reported due to the consistent trends observed in all the simulations (Fig 5); for thoroughness, the results for D1-D6 are presented in S4 Fig. The average per-residue variation is quite similar between reduced/non-reduced states. Helical regions have low variation, as expected in areas of stable secondary structure (the mode of the secondary structure on a

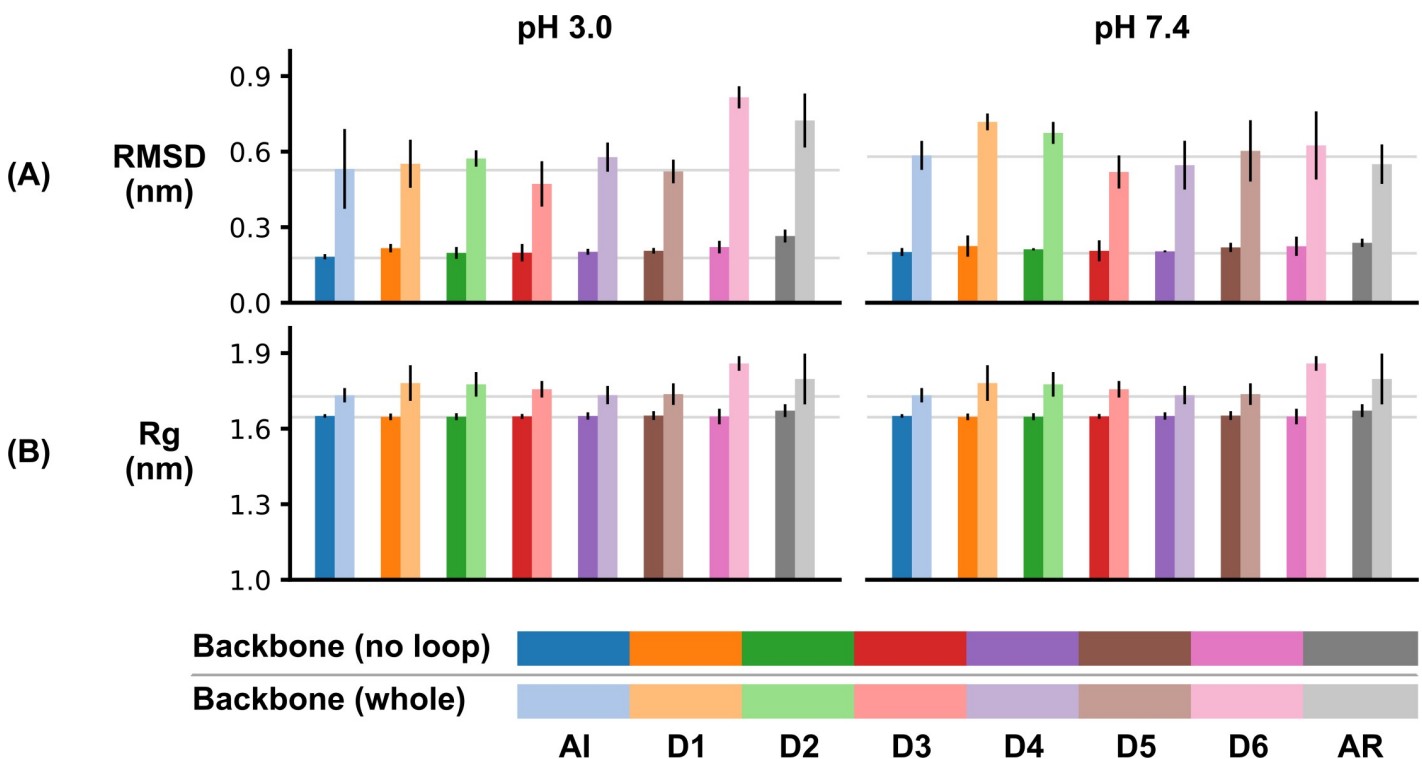

**Fig 4. Structural calculations over the last 10 ns of simulations.** RMSD (A) and $R_g$ (B) of the StPSI dimer at pH 3.0 (left) and 7.4 (right). Data presented are the average ± standard error of the mean ($n$ = 3). To enable easier parsing of the results, two grey lines have been plotted, both using the AI values for reference. The higher line is plotted at the height of the AI "whole" calculation, and the lower line is plotted at the height of the AI "no-loop" calculation.

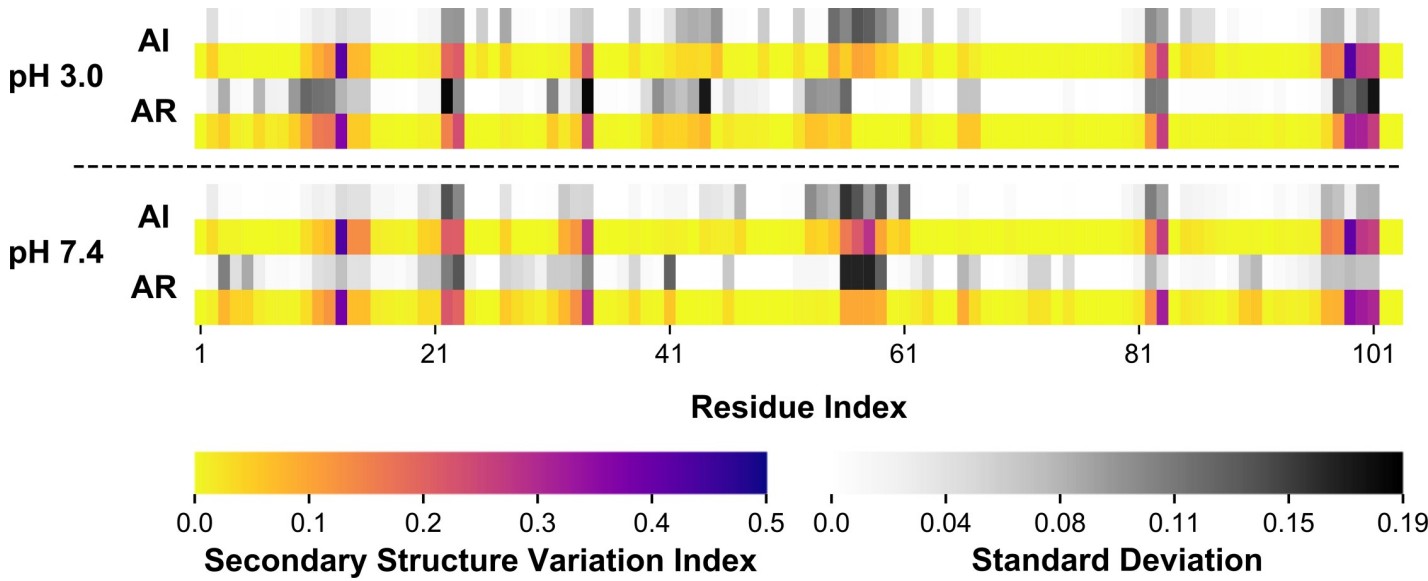

**Fig 5. Secondary structure variation of the *St*PSI dimer.** Data for AI and AR are reported for pH 3.0 (A) and pH 7.4 (B). The color of the cells is proportional to the average normalized variation per residue. The standard deviation is presented immediately above the presented variation on a separate color scale.

residue-basis can be reviewed in S5 Fig). Similarly, the loop has consistently low variation, being in a constantly unstructured state, with a low occurrence of small, short-lived beta-type or $3_{10}$ helices forming transiently. With all disulfide bonds removed, there is an increase in variation, likely the result of helix-unstructured transitions, occurring near the start of helix 1 (~residue 3). Similar slight increases can be observed mid-helix 2, 3, and 4, at both pH states. The parts of the sequence displaying the most variation are regions between helical domains. Most interestingly, in these areas, there are regions with more variation in either AI or AR, relative to each other. At pH 7.4, in the AI simulation, there is higher variation in Tyr13, and then increased variation in the C-terminal direction, whereas in the complimentary AR simulation, there is less Tyr13 variation, and instead increased variation in the N-terminal direction. Similarly, the loop-helix 4 junction is also more variable in AI than in AR. In AI, the variation is spread over nine residues and skewed towards the N-terminus, with a total average variation of 1.016 (0.112 variation per residue), whereas in AR, the variation is spread over four residues and totals 0.384 (0.096 variation per residue). Looking at pH 3.0, this asymmetric spread in variation at the same junction is even more evident, although less extreme. In the AI simulation, the increased variation is between residues 55 to 60, and in the AR simulations between residues 52 and 56. Lastly, the C-terminus was examined. At both pHs in the AI simulations, there is intense variation at Cys99, with two residues of lower variation flanking towards the end of the sequence, and two residues with lower yet variation towards the N-terminus. Conversely, in the AR simulations, there is substantially lower variation in Cys99, and diminished variation, particularly in the two residues towards the N-terminus. The implications of these findings will be further discussed in the discussion.

Regardless of the measure, there is no clear trend in any of the protein stability measurements with the disulfide bonds removed, indicating that the *St*PSI is readily stable in solution.

## Dimer interactions

Three types of non-bonded interactions were assayed to probe the *St*PSI's stability, which in part may be due to an extensive network of hydrogen bonds and hydrophobic interactions

within and between monomeric units, respectively (Fig 6). In simulations with all disulfide bonds intact, there are an average of 77 hydrogen bonds between residues within a given monomeric unit, varying slightly with pH (Fig 6). This range is similar regardless of the state of the disulfide bonds, fluctuating slightly between 72 to 77. Additionally, on average, another one to six hydrogen bonds were identified between monomeric units, for a total of approximately 150 to 160 per dimer. It is interesting to note that the AR simulations do not inherently have a lower count than the other simulations, despite their complete lack of native DSBs. The low count of hydrogen bonds between monomeric units is likely due to the hydrophobic nature of the dimer's interface. A total of 102 hydrophobic interactions were observed to occur between monomeric units of the *St*PSI dimer at pH 3.0, with all disulfide bonds intact. Side-chain interactions of wholly hydrophobic sidechains were evaluated (*e.g.*, interactions involving structures such as the aliphatic portion of the lysine sidechain were not taken into account). Most of the hydrophobic residues are localized to the inner surface of the dimer, there are fewer intra-monomer interactions–an average of 24 per monomer for the AI simulations. The number of intra-monomer hydrophobic interactions varied marginally, ranging from 23 to 25 among the remaining simulations. Essentially the same hydrophobic interaction landscape was observed at pH 7.4, with the exception of a peak count of 106 inter-monomer interactions in the D3 simulation.

Further supporting the stability of the *St*PSI are a small collection of salt bridges between residues within a given monomeric unit. Generally, there are two to four more salt bridges for a given set of simulations for pH 7.4 than pH 3.0. This can be attributed to there being less charged residues available to form salt bridges due to charge neutralization at lower pH [9]. An average of less than two inter-monomer salt bridges were found per disulfide reduced set of simulations. In these cases, acidic or basic loop residues of one monomeric unit were found to be transiently interacting with helical regions of the other monomeric unit. A similar analysis was performed to identify potential cation-π interactions, however, no physically feasible interactions were identified.

As is evident in the RMSF profile (Fig 3), with-loop RMSD (Fig 4A), and secondary structure mode (S5 Fig), the loop (residues 40–63 inclusive) is highly stochastic. We extended our analysis to specifically investigate the interactions involving the loops (Table 1). Based on the analysis, intra-loop interactions are primarily stabilized by hydrogen bonds, which range as low as 5.86 ± 2.36 (pH 3.0, D4 simulation), to as high as 9.41 ± 3.25 (pH 7.4, AI simulation). Similarly, the interactions between the loops and the rest of the dimer are also mainly through hydrogen bonds, albeit over a slightly lower range of 3.96 ± 1.98 to 7.50 ± 2.42 across all simulations. Salt bridges were uncommon and likely transient. An average of less than one salt bridge was found either intra-loop or between the loop and non-loop structures. Due to the $pK_a$ of the loop's acidic residues (Asp40, Glu54, Glu56, and Glu58) being higher than 3.0, the carboxy sidechains are protonated in the pH 3.0 simulations. In these simulations, salt bridge formation is not possible. As the loop is on the outer surface of the *St*PSI's structure, hydrophobic interactions were sparsely found. Similarly to the salt bridge interactions, an average of less than one hydrophobic interaction was found for intra-loop or loop-non-loop analyses.

## Discussion

Over the past decades, disulfide reduction has been explored peripherally alongside other experiments on saposin and saposin-like proteins [4, 5, 7, 19–22]. When examined alongside these results, new insights are shed on their possible role in this diverse family of membrane-active proteins.

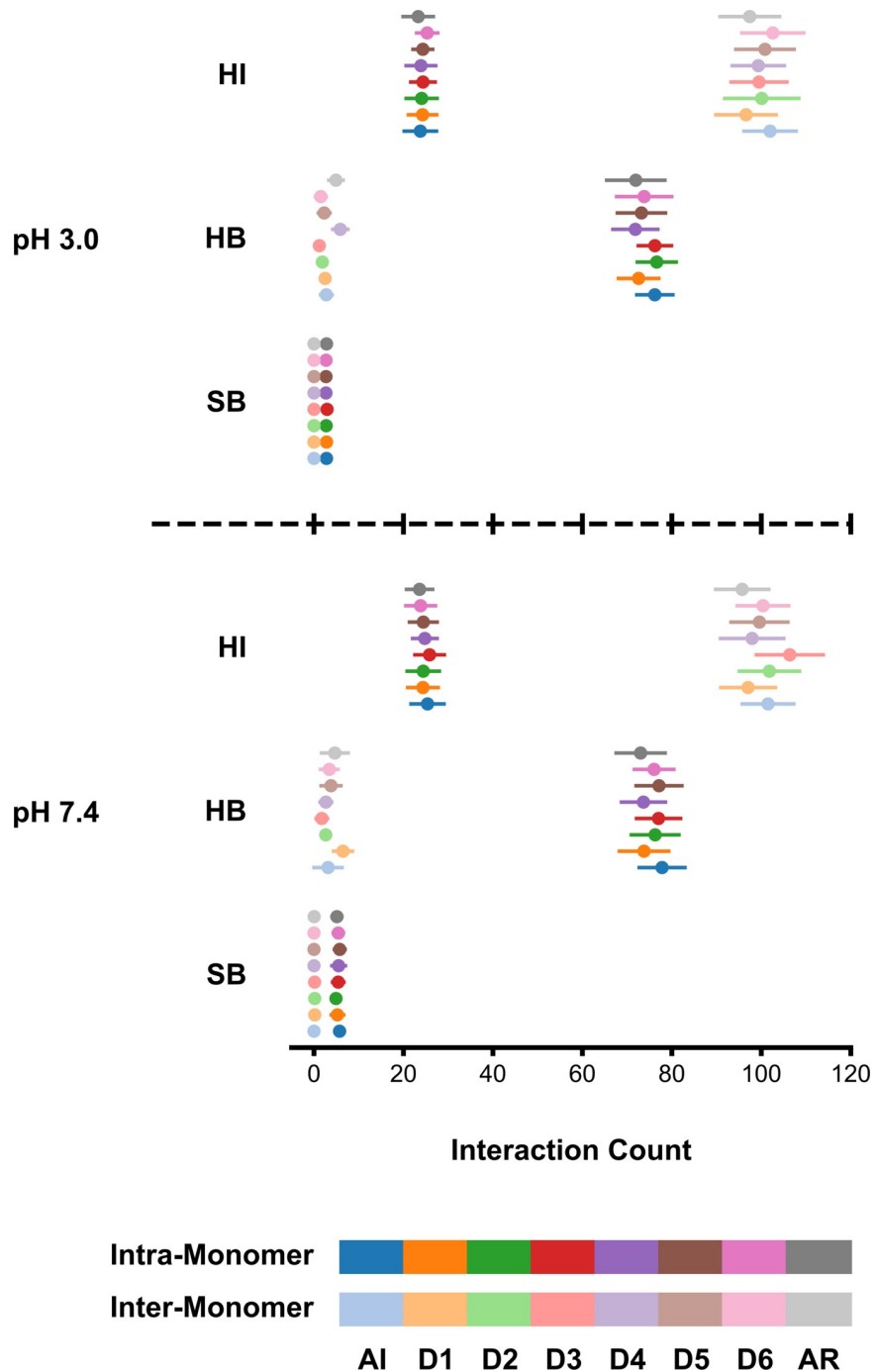

**Fig 6. Counts of hydrophobic interactions (HI), hydrogen bonds (HB), and salt bridges (SB).** pH 3.0 and 7.4 are displayed on the top and bottom, respectively. The disulfide state is indicated by the legend at the bottom of the figure. Inter-monomer data is presented as the average of the three replicates ($n = 3$), whereas intra-monomer data is presented as an average taken across both of the monomeric units for each of the three replicates ($n = 6$). The last 10% of the trajectories was utilized for the analyses. Error bars are ± standard deviation.

The structural stability of the *St*PSI was analyzed as a function of pH and disulfide bond removal. The helical domains are remarkably stable, regardless of the presence of DSBs, at either pH, which was apparent from the RMSF profile. This is consistent with previous MD

**Table 1. Counts of hydrophobic interactions (HI), hydrogen bonds (HB), and salt bridges (SB) relative to the loop portion**[*].

| | | pH 3.0 | | pH 7.4 | |
|---|---|---|---|---|---|
| | | Intra Loop | Loop-Helix | Intra Loop | Loop-Helix |
| HI | AI | 0.53 ± 0.77 | 0.04 ± 0.21 | 0.47 ± 0.69 | 0.48 ± 0.70 |
| | D1 | 0.12 ± 0.32 | 0.36 ± 0.67 | 0.49 ± 0.54 | 0.01 ± 0.12 |
| | D2 | 0.16 ± 0.37 | 0.41 ± 0.74 | 0.35 ± 0.48 | 0.33 ± 0.60 |
| | D3 | 0.39 ± 0.60 | 0.37 ± 0.73 | 0.59 ± 0.71 | 0.38 ± 0.64 |
| | D4 | 0.13 ± 0.33 | 0.32 ± 0.55 | 0.48 ± 0.72 | 0.43 ± 0.71 |
| | D5 | 0.36 ± 0.60 | 0.46 ± 0.64 | 0.28 ± 0.45 | 0.14 ± 0.36 |
| | D6 | 0.36 ± 0.78 | 0.46 ± 0.72 | 0.51 ± 0.75 | 0.14 ± 0.36 |
| | AR | 0.28 ± 0.54 | 0.49 ± 0.65 | 0.22 ± 0.53 | 0.37 ± 0.67 |
| HB | AI | 8.81 ± 2.75 | 4.96 ± 2.86 | 9.41 ± 3.25 | 6.30 ± 2.47 |
| | D1 | 6.72 ± 3.01 | 4.79 ± 3.00 | 7.71 ± 3.66 | 7.24 ± 3.94 |
| | D2 | 8.97 ± 2.20 | 4.79 ± 2.18 | 8.71 ± 3.21 | 6.05 ± 3.25 |
| | D3 | 7.44 ± 3.51 | 5.76 ± 2.95 | 9.07 ± 2.94 | 5.55 ± 1.83 |
| | D4 | 5.86 ± 2.36 | 4.29 ± 2.04 | 8.00 ± 2.70 | 5.62 ± 2.50 |
| | D5 | 5.93 ± 2.86 | 6.89 ± 2.85 | 8.73 ± 3.35 | 7.50 ± 2.42 |
| | D6 | 8.28 ± 3.50 | 3.96 ± 1.98 | 9.37 ± 3.31 | 5.87 ± 2.81 |
| | AR | 7.75 ± 2.62 | 4.52 ± 2.41 | 8.89 ± 2.46 | 4.47 ± 2.55 |
| SB | AI | n.d. | 0.00 ± 0.05 | 0.00 ± 0.00 | 0.03 ± 0.16 |
| | D1 | n.d. | 0.11 ± 0.31 | 0.00 ± 0.00 | 0.28 ± 0.45 |
| | D2 | n.d. | 0.00 ± 0.02 | 0.00 ± 0.00 | 0.19 ± 0.40 |
| | D3 | n.d. | 0.06 ± 0.24 | 0.00 ± 0.00 | 0.29 ± 0.46 |
| | D4 | n.d. | 0.01 ± 0.10 | 0.00 ± 0.00 | 0.23 ± 0.44 |
| | D5 | n.d. | 0.17 ± 0.38 | 0.00 ± 0.00 | 0.08 ± 0.28 |
| | D6 | n.d. | 0.13 ± 0.34 | 0.00 ± 0.00 | 0.48 ± 0.58 |
| | AR | n.d. | 0.17 ± 0.38 | 0.00 ± 0.00 | 0.14 ± 0.34 |

[*]The data is presented as an average taken across both of the monomeric units for each of the three replicates ($n = 6$). The last 10% of the trajectories was utilized for the analyses. Error is standard deviation.

simulations on the *St*PSI with all disulfide bonds intact [8]. The majority of the fluctuation in the profile can be attributed to the loop portion, being largely disordered with the exception of small, transient $\alpha/3_{10}$ helical sections or small β-bridges (S5 Fig). This is further evidenced by the low loop-excluded RMSD, which is explained by the high number of intra-monomer hydrogen bonds that reinforce and stabilize the helical domains. Results seen in this study were consistent with those reported in previous studies regarding RMSD values with inclusion of the loop portion that are stable over later portions of the trajectory [8]. Despite the chaotic behaviour, the loop portion is likely closely associated with the helical domains. The radius of gyration varies little and has a small range: 1.7252 to 1.8589 nm for with-loop and 1.6314 to 1.6716 nm for without-loop calculations. Thus the loop is likely in close contact to the dimer's helical bundle, but unrestricted, which was further supported by the high with-loop RMSD.

Based on the above results, the disulfide bonds have no apparent structural role in a state lacking influence from other extrinsic factors (*e.g.*, membranes, cellular components, *etc.*). RMSF, RMSD, and $R_g$ all indicate that the *St*PSI is readily stable, regardless of pH or disulfide-reduction state. Bryksa *et al.* (2011) also suggested that the disulfides are likely not required for preserving the general saposin-like protein fold [4]. Instead of the disulfide bonds giving the protein rigidity when interacting with membrane surfaces and excising/maneuvering lipids, they may have implications for specificity. Therefore, the disulfide bonds may become

important for other functions of the PSI, such as targeting the parent aspartic protease zymogen to the central vacuole.

Generally, it had been assumed that disulfide bonds were required for the saposins or other saposin-like proteins to perform their *in vivo* roles properly [3]. Saposin C with all disulfide bonds removed and blocked with carboxymethyl groups was unable to activate glucosylceramidase or fuse vesicles [6]. However, this may be due to an altered electronic footprint contribution from the carboxymethyl groups [6]. Conversely, when saposin C cysteine mutants were expressed such that one bonding participant per disulfide bond was reduced activity was observed. In Cys47Ser, Cys72Gly, Cys72Trp, and Cys78Arg, all were capable of activating a glucosylceramidase enzyme and retained native heat stability [22]. In saposin B, reduction and subsequent blocking of the sulfhydryl groups with vinylpyridine yielded protein products capable of potentiating cerebroside hydrolysis [7] although reduced-blocked proteins were more susceptible to enzymatic digestion via trypsin [7].

Prosaposin is targeted to lysosomes [23], and the derived saposins function in this lytic environment [24]. Similarly, the PSI targets the parent AP to the central vacuole [25], which is acidified and akin to the mammalian lysosome [26, 27]. The functional requirement of low pH [28] is also consistent with function inside the vacuole. By maintaining a tightly packed hydrophobic core, extraneous hydrolysis of the protein due to pH or incidental vacuolar protease activity may be prevented, thus theoretically extending the useable lifetime of the PSI *in vivo*. The loop portion, being relatively highly mobile, would not be protected in this theory. However, this may be of little consequence to function (*e.g.*, aspartic protease targeting). The saposins are fully functional without a large loop portion, indicating that it may be dispensable. Furthermore, the PSIs helices flanking the loop region would be securely anchored together by the two disulfide bonds found in this region (in the *St*PSI, Cys31-Cys71 and Cys37-Cys68) despite lacking a loop portion. The latter assumes the preservation of the PSI in general after excision from the parent AP, however, the fate of the PSI post-targeting is unknown [29], and addresses the importance of the large number of hydrophobic interactions identified–up to 106 between monomeric units–which likely stabilizes the dimer interface. Similar dimer forms are seen in saposins A-D [18, 30–32].

In previous studies, *in vitro* DTT reduction led only to minor qualitative losses in helicity at both acidic and neutral pH in the *St*PSI [4, 33]. Reduced saposin B was observed to have a higher helix content after reduction (63.9% compared to 51.2% in the native protein) [21], however, the effects of purely disulfide bond reduction cannot be delineated from structural perturbation from the modification procedure. In this work, the secondary structure is instead reported as variation, or change in secondary structure, and was observed to fluctuate slightly throughout the protein sequences. For example, in Fig 5, there is increased variation about Tyr13 in the AR simulation at pH 3.0, however, there is also relatively less variation at the end of helix 4, compared to the AI simulations at pH 3.0. When re-calculated as a sum over the entire dimer, net variation remains relatively constant across the simulations (S1 Table). Saposin B has been the subject of hydrogen-deuterium exchange experiments performed with and without the three native disulfide bonds intact [21]. In this study, experiments on native saposin B yielded a total exchange of 77 to 80 protons, whereas disulfide reduced saposin B exchanged ~125 (out of a total of 134 and 140 total theoretically exchangeable protons, respectively). In addition to having a larger population of exchangeable protons, the disulfide reduced saposin B also exchanged them much faster. With the disulfide bonds removed, saposin B was able to access new conformations, but still maintained the motifs necessary to activate the parent enzyme arylsulfatase A. Using these findings as a guide, the *St*PSI is likely similarly affected. With the removal of the *St*PSI's disulfide bonds, minor changes occur in the underlying microstructure, thereby allowing access to new conformations, which has

previously been theorized to allow for enhanced membrane interactions [33]. Disulfide removal in other lipid-active proteins, such as human tear lipocalin, led to faster dynamics due to enhanced flexibility of structural features [34], thus, the disulfides may confer membrane specificity to the PSI, more so than govern actual activity. In four unique PSIs, each displayed an enhanced ability to induce vesicle size changes relative to their respective controls upon disulfide reduction [5].

Visually, the changes in secondary structure variation are minor (Figs 5 and S4). This is further supported by the normalized dimer variation being similar across all simulations with low standard deviation (S1 Table), and the mode of the secondary structure being consistent across simulations (S5 Fig). Thus, the net secondary structure does not change, instead, some regions with low variation are destabilized, accompanied by a concomitant decrease in areas of the structure that initially had higher variation. Upon removal of one/two/three disulfide bonds, various regions are able to release structural frustration and adopt slightly varied secondary structure, as evidenced by the different secondary structure variation profiles (Figs 5 and S3), which may in turn lead to slightly perturbed tertiary and quaternary structures (S6 Fig). This was observed in a saposin-like protein from *Fasciola hepatica* [35]. Upon disulfide bond reduction, overall secondary structure was maintained, but some conformational epitopes were lost, as evidenced by attenuated binding to rabbit sera, *i.e.*, the interaction landscape was restructured. This is observed with the *St*PSI as well, through the small changes observed in secondary structure variation (Figs 5 and S4) and differential hydrogen bond, salt bridge, and hydrophobic interactions (Fig 6) as a function of disulfide bond reduction. These differences have the potential to lead to more polar or hydrophobic residue exposure, which can then differentially enhance interactions phospholipid headgroups or lipid tails, respectively. In the *St*PSI, these effects are realized as enhanced membrane interactions [5] and a stable net secondary structure [4] in the absence of disulfide bonds. Similarly function in saposins B [7] and C [22] is maintained when disulfide bonds are removed, either by mutation or derivatization.

The notion that the disulfide bonds guide membrane specificity [5] and have a minimal role in maintaining structure [33] is a reasonable conclusion given the existence of many saposin-like proteins lacking disulfide bonds, and is further supported by the results presented in this work. Such functional examples are found outside of *Eukarya* and include: a secretion factor in *Vibrio vulnificus* (*Prokarya*) [36], a putative calcium-binding protein in *Sulfolobus solfataricus* (a hyperthermophile in *Archaea*) [37], various cyclic bacteriocins [38, 39], and other small prokaryotic cyclic peptides [40–42]. See S7 Fig for a structural alignment of some of these peptides, relative to the native monomer of the barley PSI.

In summary, the present work indicates that disulfide bonds at acidic or neutral pH have a minimal role in the structural stability of the StPSI in solution. Unique findings include that the *St*PSI dimer is stabilized predominantly by hydrophobic interactions at the shared interface of the monomeric units, resulting in a dimer structure similar to saposin C, and that a higher degree of structural plasticity is adopted upon disulfide reduction, leading to an ability for residues to slightly alter the secondary structure in specific regions. This may relieve stress and allow for the adoption of slightly altered secondary or tertiary structures conducive to enhanced membrane interactions that have been observed *in vitro*.

## Methods

### Protein structure preparation

The starting structure was taken from [4]; unresolved residues (Asp40-Gly63) were rebuilt using MODELLER [43] as random coil using Chimera version 1.11.2 [44]. Cys-Cys linkages are present *in vivo* and connect Cys6-Cys99, Cys31-Cys71, and Cys37-Cys68. Simulations

were performed with the disulfide bonds removed (Table 2) singly (coded D1, D2, D3) or doubly (coded D4, D5, D6). Two control simulations were also performed (Table 2): with all the disulfide bonds intact (AI–"all intact") and with all the disulfide bonds reduced (AR–"all reduced"). Charges to approximate protonation states at pH 3.0 and 7.4 were assigned using previously calculated $pK_a$ values [9].

## All-atom molecular dynamics

All-atom simulations were performed as per our previous publication [9] using Gromacs [45] with the AMBER99SB-ILDNP forcefield [46–49]. This forcefield was selected as it has side-chain torsion potentials benchmarked against quantum mechanics calculations and NMR experiments. Gromacs version 5.0.4 was used for simulation preparation and analysis; Gromacs version 5.0.7 was used for performing production MD simulations on Compute Canada's high-performance computing cluster Cedar and Graham. Starting with the structures outlined in Section 4.1, simulation systems were solvated using the TIP3P water model and ions added to 150 mM NaCl, plus neutralizing ions. Minimization was performed, followed by a brief (1 ps) isochoric simulation to initiate the system using a "cutoff" scheme for van der Waals interactions. Next, using the particle-mesh Ewald approach [50] to calculate both van der Waals and columbic interactions, sequential execution of isochoric and isobaric equilibrations for 500 ps with a 2 fs time step were performed. Production MD simulations were carried out for 100 ns. The neighbour list was updated every 10 steps using a Verlet cutoff scheme [51]. Electrostatics were smoothly shifted to zero between 0.9 and 1.0 nm. Temperature was maintained at 310 K ($\tau_T$ = 0.1 ps) using the V-rescale algorithm [52]; pressure was maintained at 1 atm ($\tau_P$ = 2 ps, compressibility = 4.5 x $10^{-5}$) using Parrinello-Rahman pressure coupling [53]. Three independent replicates were performed for each pH-disulfide-reduction combination, and the AI and AR controls (Table 2).

## Analysis

Following simulations, the protein dimer was clustered and extracted for analysis. The dimer was made whole across periodic boundaries. For all analyses, further processing was performed largely using in-house Python scripts (Python version 2.7).

**Inter-sulfur distances.** The distance between pairs of cysteine sulfur atoms, Sγ, were calculated using *gmx mindist*. The cysteine pairs analyzed were those that natively form disulfide bonds. The data was histogrammed using a bin-width of 0.01 nm. As the dimer is symmetric, data for both monomers in all three replicates were pooled ($n$ = 6), with the average and

**Table 2. Disulfide reduction sample code key.**

| Disulfide Bond* | | | Sample Code |
|---|---|---|---|
| Cys6-Cys99 | Cys31-Cys71 | Cys37-Cys68 | |
| I | I | I | AI |
| R | I | I | D1 |
| I | R | I | D2 |
| I | I | R | D3 |
| R | R | I | D4 |
| I | R | R | D5 |
| R | I | R | D6 |
| R | R | R | AR |

* "R" denotes the disulfide bond is reduced, "I" denotes the disulfide bond is intact

standard deviation for each bin presented. A supplemental analysis exploring the possibility of disulfide interchange between the two closely position disulfides was also performed.

**Structural analyses.** RMSF, RMSD, and $R_g$ were calculated using built-in Gromacs programs (*gmx rmsf*, *rms*, *gyrate*).

**Secondary structure variation.** Secondary structure analysis over the last 10 ns was made using DSSP (version 2.2.1) [54] performed *via* Gromacs *do_dssp*. Secondary structure elements were indexed into helical character, beta character, and disordered (coil, turn, bend). From this data, a secondary structure variation index was created. Change events between any of the secondary structure indices were summed by residue, and then normalized to the maximum theoretical value of the variation (1000, indicating a change every frame in the trajectory) to ascertain the tendency of different residues to change secondary structure.

**Contacts analysis.** To perform the contacts analysis from the point of view of different non-bonded interactions, namely hydrophobic interactions, hydrogen bonds, and salt bridges, each interaction was calculated separately over the last 10 ns of the 100 ns simulations. Hydrogen bonds were quantified using *gmx hbond* with default settings. Hydrophobic interactions were determined by creating a minimum distance contact map using only hydrophobic amino acid sidechains and a cutoff of 0.45 nm. Salt bridges were analyzed in a similar manner, with the same cutoff. The minimum distance separating sidechain atoms between acidic and basic amino acids was calculated. For acidic amino acids, only sidechain oxygen atoms were used for the calculation, with a further caveat that at pH 3.0 they also needed to be deprotonated. For basic amino acids, the nitrogen atom used for the calculation was dependant on the residue: lysine–the terminal sidechain nitrogen; arginine–the center of mass of all nitrogen atoms in the guanidinium group; histidine–the center of mass of the distributed positive charge between sidechain $\delta/\epsilon$ nitrogen atoms and associated $\epsilon$ carbon atom. Histidine was omitted from the salt bridge analysis at neutral pH due to its protonation state at pH 7.4 [9]. For each type of interaction, the count of interactions per frame was determined for each of the three replicates. The data for all replicates over the last 10 ns was then pooled, and the average and standard deviation of the data set calculated.

## Supporting information

**S1 Table. Normalized dimer variation.**
(DOCX)

**S1 Fig. Highlighting the domain-swapped nature of PSIs relative to other SAPLIPs.** Panel A) Structural alignment of the *Solanum tuberosum* PSI (*St*PSI) (PDB ID: 3RFI) and saposin C (PDB ID: 2QYP). Note: The unresolved loop section in the *St*PSI was omitted in this alignment and rendering. Coloring is as per upper legend for panel A. Panel B) Clustal-Omega alignment of the C-terminal half of four PSIs with the N-terminal half of saposins A-D. Coloring is as per bottom legend. SapA–saposin A; SapB–saposin B; SapC–saposin C; SapD–saposin D; AtPSI–*Arabidopsis thaliana* PSI; CcPSI–*Cynara cardunculus* PSI; HvPSI–*Hordeum vulgare* PSI. The loop portion of the PSIs used was omitted from the alignment. Panel C) Clustal-Omega alignment of the N-terminal half of four PSIs with the C-terminal half of saposins A-D. Details are as per Panel B.
(TIFF)

**S2 Fig. Separation between feasible non-native cysteine sulfur atom pairs at pH 3.0 and pH 7.4.** The pairs examined are between C31-C37 (blue), and C68-C71 (orange). Data were histogrammed over the last 10% of the simulations for each monomer and replicate. Dark lines are the average per-bin across the replicates and monomeric units (n = 6). Lighter shaded regions

correspond to the per-bin standard deviation. The grey dashed line denotes 0.4 nm, the cutoff used in this paper to define a possibility of disulfide interchange occurring.
(TIFF)

**S3 Fig. Backbone RMSD of 100 ns AR simulations compared to an independent AR simulation for 1 μs.** pH 3.0 –top; pH 7.4 –bottom. Red shades– 100 ns AR simulations (); blue– 1 μs AR simulation. NL–loop residues 40 to 63 omitted for the calculation; WL–loop residues included for the calculation. Bar graphs on right are the average ± standard deviation or standard error of the mean (where appropriate) over the last 10% of the simulation trajectory/trajectories.
(TIFF)

**S4 Fig. Secondary structure variation as a function of pH and disulfide reduction.** pH 3.0 –top; pH 7.4 –bottom. For sample codes, see Table 1. The color of the cells is proportional to the average normalized variation per residue. The standard deviation is presented immediately above the presented variation on a separate color scale.
(TIFF)

**S5 Fig. Secondary structure mode as a function of pH and disulfide reduction.** pH 3.0 –top; pH 7.4 –bottom. The color of the cells represents the mode of the calculated secondary structure on a residue basis.
(TIFF)

**S6 Fig. Center-of-mass separation between helical elements or monomers.** Left: The separation of helices relative to helix 1 are presented for example purposes. Right: The separation of monomer A and monomer B, with the loop portion (WL) or with no-loop (NL) portion included in the calculation.
(TIFF)

**S7 Fig. Aligned saposin-like proteins in a closed conformation.** PDB codes for each structure are given in the figure. If more than one model was present in the PDB file, only the first one model was used; *Loop residues Cys320 to Asp349 were omitted from the figure for clarity; †N-terminal residues Lys91 to His99 were omitted from the figure for clarity.
(TIFF)

# Acknowledgments

The authors would like to thank West Grid and Compute Canada for access to their respective high-performance computer clusters for running the simulations (Cedar and Graham). We sincerely thank Dr. Brian C. Bryksa and Lennie K.Y. Cheung for many helpful discussions.

# Author Contributions

**Conceptualization:** John H. Dupuis, Chen Song, Rickey Y. Yada.

**Data curation:** John H. Dupuis.

**Formal analysis:** John H. Dupuis.

**Funding acquisition:** Rickey Y. Yada.

**Investigation:** John H. Dupuis.

**Methodology:** John H. Dupuis.

**Project administration:** John H. Dupuis.

**Resources:** John H. Dupuis, Rickey Y. Yada.

**Supervision:** Chen Song, Rickey Y. Yada.

**Validation:** Chen Song.

**Visualization:** John H. Dupuis.

**Writing – original draft:** John H. Dupuis.

**Writing – review & editing:** John H. Dupuis, Shenlin Wang, Chen Song, Rickey Y. Yada.

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
