## [Decision Letter · Decision Letter 0]

4 Jun 2020

PONE-D-20-12645

The role of disulfide bonds in a Solanum tuberosum saposin-like protein investigated using molecular dynamics

PLOS ONE

Dear Dr. Yada,

Thank you for submitting your manuscript to PLOS ONE. After careful consideration, we feel that it has merit but does not fully meet PLOS ONE’s publication criteria as it currently stands. Therefore, we invite you to submit a revised version of the manuscript that addresses the points raised during the review process.

Specifically, there are a number of technical concerns raised by the reviewers that should be addressed before the manuscript becomes acceptable.

We look forward to receiving your revised manuscript.

Kind regards,

Oscar Millet

Academic Editor

PLOS ONE

Journal Requirements:

Reviewers' comments:

Reviewer's Responses to Questions

**Comments to the Author**

1. Is the manuscript technically sound, and do the data support the conclusions?

Reviewer #1: No

Reviewer #2: Yes

2. Has the statistical analysis been performed appropriately and rigorously? 

Reviewer #1: N/A

Reviewer #2: Yes

3. Have the authors made all data underlying the findings in their manuscript fully available?

Reviewer #1: Yes

Reviewer #2: Yes

4. Is the manuscript presented in an intelligible fashion and written in standard English?

Reviewer #1: Yes

Reviewer #2: Yes

5. Review Comments to the Author

Reviewer #1: The authors have used molecular dynamics to study the effects have on tridimensional structure of the removal of the disulfide bonds of the St plant specific insert (StPSI). The main result from this work is that the removal of the disulfide bonds do not significantly affect the protein secondary structure and have a minimal role on protein stability. I found the work interesting but also preliminary.

I have several questions and comments:

1. If disulphide bonds in the protein are thought to stabilize it, why when removing them does the structure remain practically the same?. Could it be possible that 100ns were too few to perform the simulation?. I would increase the simulation time to at least 400ns to see if the protein structure changes. 100ns is too little. If you run the simulation for 400ns, you could analyse the data for the last 40ns (10% of simulation time). Check also SF2 (In this case 1us simulation time for AR is compared with 100ns simulations).

2. Page 5, lines 76-78. Rewrite. I do not understand the phrase.

3. Page 7, lines 119-130, Figure 2. Why not use histograms of the last 10% of the simulation?. Comparisons would be easier.

4. Rearrange Figure 2 so that the grey plots are removed.

5. According to the legend of Figure 2, is it true that the distances of intact disulphide bonds are fixed?. The protein should not be fixed at any moment, even if you remove one disulphide bond.

6. Page 10, line 208. I do not get it.

7. Page 14, Protein topology. The CAST study does not add anything new and/or interesting. Delete.

8. Remove Section 4.3.6, Protein topology characterization.

9. Two and a half pages for Introduction, seven pages for Results and six pages for Discussion ?. I suggest to significantly reduce the Discussion section.

10. Delete the last two phrases of the discussion section (page 22, lines 436-440). Lucubration without data to support it.

Figures.

1. Delete Part A, Figure 1.

2. Delete Figures SF5 and SF6.

3. Delete Table 1.

Reviewer #2: Report on The role of disulfide bonds in a Solanum tuberosum saposin-like protein investigated using molecular dynamics

This is a well-written manuscript describing simulations of a protein of interest to saposin-superfamily researchers as well as to those in the field of plant defensive systems. The simulations seem to be robustly carried out and competently analyzed. The trajectories were available online, which was super. One complaint, that pertains only to the review process, is that the figures, figure legends, and text pertaining to the figures wound up in 3 completely different places in the pdf file. This meant that it was necessary to have 3 different windows open at the same time to read through the manuscript which was inconvenient.

The following are minor suggestions to improve the clarity of the presentation.

Line 47: “These unique findings…”. I don’t think the findings are “unique” as the non-covalent bonds that stabilize saposin superfamily proteins have been studied before.

Line 58: “PSIs are domain swapped with respect to other SAPLIPs”. I’m not sure what this means. Can you lengthen the sentence to give a bit more info

Line 66: Caption for figure 1. Can you add more info so that readers don’t have to track back through the methods and then the references to figure out what they’re looking at. I.e. is this an experimental structure that was used as the starting model for the simulations?

Line 91: I’m not sure “motivators” is the right word here.

Lines 95-98. I don’t think it’s necessary to justify MD as a useful technique. I would remove these lines and replace them with a few sentences on previous MD work with saposin superfamily proteins.

Line 101: again so we don’t have to track back through the referenced paper – was reference 11 an experimental structure or an MD simulation?

Line 121-122: I’m not sure what “complementary to removal of these disulfide bonds singularly” means – I suggest rewording.

Line 137 and figure 2: refers to “shaded region above and below curves in Fig 2”. I can’t see a shaded region. Clarify.

Line 169 – tell me which simulation (D_) I should be looking at (so that I don’t have to refer back to the supplementary table to find which panel in figure 3 to look at.

Line 187-188, “the maxima and minima are for the AI and AR simulations, respectively”. Are you sure? I can’t see any evidence for this. Can you clarify what I should be looking at.

Line 283 – typo – “report” should be “reports”

Line 313 – I think it would be helpful to tell us more about the interactions that stabilize the association between the loop portion and the helical domains (i.e. are they hydrophobic, hydrogen bonds, involving specific pairs of residues, etc).

Line 317 – what is the “unperturbed state” being referred to?

Line 341 – did you see any differences in the simulations at the two pHs?

Line 348-349 – re-write to avoid the use of two “however”s in one sentence.

Line 373 – typo “In” should be “in”

Line 395 and 420 – what type of “specificity” is being referred to? Specificity for a particular membrane composition? Or substrate?

Line 432 – does this work really tell you about “function”? I think this needs to be softened – maybe just to “potential” function.

Line 457. I would have appreciated one sentence – either here or in the results – explaining your main reasons for picking this particular force field.

General… it would be nice to have a sequence alignment (probably in the supplementary figures) that shows the alignment between StPSI and the other saposins referred to throughout the text – e.g. Saposin C, etc.”

6. PLOS authors have the option to publish the peer review history of their article (what does this mean?). If published, this will include your full peer review and any attached files.

Reviewer #1: No

Reviewer #2: Yes: Valerie Booth

---

## [Author Response · Author response to Decision Letter 0]

15 Jul 2020

Please note: 

All responses are tracked in BLUE text in the docx, and indicated with "Reponse:"

Line numbers provided are relative to the copy of the manuscript with tracked changes in full-markup. 

References in sections of text that were deleted were not tracked. This was necessary for the reference manager to track references properly.

Please note: Two figures were included in the responses to reviewers, which are only available in the docx file uploaded.

Dear Dr. Yada,

Thank you for submitting your manuscript to PLOS ONE. After careful consideration, we feel that it has merit but does not fully meet PLOS ONE’s publication criteria as it currently stands. Therefore, we invite you to submit a revised version of the manuscript that addresses the points raised during the review process.

Specifically, there are a number of technical concerns raised by the reviewers that should be addressed before the manuscript becomes acceptable.

Response: We have updated the manuscript to conform to these guides.

Response: We have linked the corresponding author’s ORCID to this submission.

Reviewers' comments:

Reviewer's Responses to Questions

Comments to the Author

1. Is the manuscript technically sound, and do the data support the conclusions?

Reviewer #1: No

Reviewer #2: Yes

2. Has the statistical analysis been performed appropriately and rigorously? 

Reviewer #1: N/A

Reviewer #2: Yes

3. Have the authors made all data underlying the findings in their manuscript fully available?

Reviewer #1: Yes

Reviewer #2: Yes

4. Is the manuscript presented in an intelligible fashion and written in standard English?

Reviewer #1: Yes

Reviewer #2: Yes

5. Review Comments to the Author

Reviewer #1: The authors have used molecular dynamics to study the effects have on tridimensional structure of the removal of the disulfide bonds of the St plant specific insert (StPSI). The main result from this work is that the removal of the disulfide bonds do not significantly affect the protein secondary structure and have a minimal role on protein stability. I found the work interesting but also preliminary.

I have several questions and comments:

1. If disulphide bonds in the protein are thought to stabilize it, why when removing them does the structure remain practically the same?. Could it be possible that 100ns were too few to perform the simulation?. I would increase the simulation time to at least 400ns to see if the protein structure changes. 100ns is too little. If you run the simulation for 400ns, you could analyse the data for the last 40ns (10% of simulation time). Check also SF2 (In this case 1us simulation time for AR is compared with 100ns simulations).

Response: Although disulfide bonds are thought to stabilize proteins in general, we make no claim in the manuscript that disulfide bonds are crucial for stabilizing the StPSI’s structure. Instead, we highlight this previously held notion in the Discussion, clarifying that it was previously thought that disulfide bonds were required for function (“Generally, it had been assumed that disulfide bonds were required for the saposins or other saposin-like proteins to perform their in vivo roles properly [3]”) (L387-388). 

This manuscript was written from the point of view that removing disulfide bonds results in several interesting properties based on previous experiments done on PSIs/saposins (increased trypsin sensitivity, no changes in secondary structure, increased membrane interactions). We specifically avoided comments about the disulfide bonds stabilizing the structure as there is ample experimental evidence that saposins/saposin-like proteins are stable without their disulfide bonds (some references supporting this claim can be found in references 4, 5, 7, 21, & 22 in the manuscript), which were supported by our simulation results.

We agree that longer simulations may have been warranted, however, based on our 1 μs control simulations (using the AR configuration with all disulfide bonds removed), the 100 ns simulations appeared to be sufficient. 

In the last 10% of the 1 μs simulation, the backbone RMSD of the dimer (excluding the loop portion) was 0.210 ± 0.014 and 0.252 ± 0.015 nm, for pH 3.0 and 7.4, respectively (the error is the standard deviation). For the last 10% of the 100 ns simulation replicates, the backbone RMSD of the dimer (excluding the loop portion) was 0.265 ± 0.026 and 0.238 ± 0.016, for pH 3.0 and 7.4, respectively (the error is the standard error of the mean). This data is visually presented in S3 Fig (bar chart at left). The fact that these two sets of averages are so similar indicates to us that structurally, the StPSI is unlikely to undergo large conformational changes, even on a long time-scale. 

We have provided a figure (Figure R1, see at the end of the responses) of the Cα separation between Cα atoms of disulfide bonding partners (C6-C99, C31-C71, and C37-C68). As can be seen in the figure, the separation distribution is very similar between the last 10% of the 1 μs simulation and the last 10% of three 100 ns replicates. The fact that these two sets of data are very similar indicates convergence.

We also provide a second figure (Figure R2) of the center-of-mass separation between monomers, and between all helical elements in all combinations. The last 10% of the 1 μs simulation was compared to the last 10% of the three 100 ns simulations. Inter-monomer separations are again very similar between the two simulation lengths, regardless of whether or not the loop section was included. Examining the inter-helical separations, many of the helices have similar separations between the two simulations.

The above three structural characterizations indicate that there is not an appreciable difference between the outcome of the three 100 ns simulations compared to the outcome of longer 1 μs simulations. Given the above, we feel that the three 100 ns simulation replicates performed in the manuscript have adequately captured the structural dynamics that would be expected to be observed on a longer time-scale.

2. Page 5, lines 76-78. Rewrite. I do not understand the phrase.

Response: This sentence has been revised to improve clarity. The PE/PS abbreviation has also been removed as it is not used in the rest of the manuscript.

Original:

“In all four cases, vesicle size increases in 1:1 16:0-18:1 phosphatidylethanolamine/16:0-18:1 phosphatidylserine (PE/PS) vesicles at pH 4.5 were greater after reduction of disulfides than with intact disulfides [4], with similar results seen in saposin C [5].”

Revision: 

“Earlier works have been carried out to study the relationship between the PSI’s vesicle fusion activity and the presence of disulfide bonds [5]. Using a common experimental platform (1:1 16:0-18:1 phosphatidylethanolamine/16:0-18:1 phosphatidylserine vesicles at pH 4.5), PSI-induced vesicle size increases were monitored as a function of disulfide-state [5]. For each of the four aforementioned PSIs, the ability of the PSI to increase vesicle size improved after reducing the disulfide bonds, with respect to those with intact disulfide bonds. These experimental results were also similar to those seen with saposin C [6]”

3. Page 7, lines 119-130, Figure 2. Why not use histograms of the last 10% of the simulation?. Comparisons would be easier.

Response: Thank you for this suggestion. The data was re-plotted as a histogram, using the last 10% of each simulation to produce histograms. The results, discussion, and methods have been updated to reflect this change. S2 Fig (disulfide exchange) was also updated in the same manner.

4. Rearrange Figure 2 so that the grey plots are removed.

Response: Figure 2 has been replotted as per the above comment, therefore, the filler plots are no longer necessary.

5. According to the legend of Figure 2, is it true that the distances of intact disulphide bonds are fixed?. The protein should not be fixed at any moment, even if you remove one disulphide bond.

Response: We apologize for the misleading wording. No part of the protein was fixed at any time during these production MD simulations. In the Amber force field used in this work (amber99sb-ildnp), the equilibrium distance between sulfur atoms in an intact disulfide bond is held [nearly] constant at 0.2038 nm using a bond with a spring constant of 138,908.8 kJ mol-1 nm-2. Thus, although not fixed, this value deviates little in intact disulfide bonds. In the “all-intact” simulations, the bond length of the intact disulfide bonds, pooled, was 0.203,800,2 ± 0.000,003,9 nm (n = 180,018). There were no distance restraints between the sulfur atoms after removing the disulfide bond. The legend for Figure 2 has been revised in response to comment #3, and as a result, the term “fixed” has been omitted.

6. Page 10, line 208. I do not get it.

Response: “For posterity” has been revised to “For thoroughness”

7. Page 14, Protein topology. The CAST study does not add anything new and/or interesting. Delete.

8. Remove Section 4.3.6, Protein topology characterization.

Response (to 7 and 8): We have removed the topological characterization from the manuscript.

9. Two and a half pages for Introduction, seven pages for Results and six pages for Discussion ?. I suggest to significantly reduce the Discussion section.

Response: In response to comments about the (lack of) necessity for topological characterization, approximately 1 page of discussion has been removed.

10. Delete the last two phrases of the discussion section (page 22, lines 436-440). Lucubration without data to support it.

Response: These phrases have been removed. 

Figures.

1. Delete Part A, Figure 1.

Response: Figure 1A indicates the symmetrical nature of the dimeric structure, as well as the model used in our simulations. Therefore, we believe that Figure 1A is necessary.

2. Delete Figures SF5 and SF6.

Response: We have removed these supplemental figures

3. Delete Table 1.

Response: Table 1 in the original manuscript has been removed. Note: There is still a “Table 1” in the revised manuscript since Table S1 is now part of the main manuscript.

Reviewer #2: Report on The role of disulfide bonds in a Solanum tuberosum saposin-like protein investigated using molecular dynamics

This is a well-written manuscript describing simulations of a protein of interest to saposin-superfamily researchers as well as to those in the field of plant defensive systems. The simulations seem to be robustly carried out and competently analyzed. The trajectories were available online, which was super. One complaint, that pertains only to the review process, is that the figures, figure legends, and text pertaining to the figures wound up in 3 completely different places in the pdf file. This meant that it was necessary to have 3 different windows open at the same time to read through the manuscript which was inconvenient.

Response: We realize this is not ideal, however, the manuscript was formatted as per the PLOS submission guidelines, which indicate figures are to be preferentially submitted as image files, and not in the main manuscript.

The following are minor suggestions to improve the clarity of the presentation.

Line 47: “These unique findings…”. I don’t think the findings are “unique” as the non-covalent bonds that stabilize saposin superfamily proteins have been studied before.

Response: The word “unique” has been removed. 

Line 58: “PSIs are domain swapped with respect to other SAPLIPs”. I’m not sure what this means. Can you lengthen the sentence to give a bit more info

Response: An explanation for “domain swapped” has been added at L59-61, and in S1 Fig. 

Line 66: Caption for figure 1. Can you add more info so that readers don’t have to track back through the methods and then the references to figure out what they’re looking at. I.e. is this an experimental structure that was used as the starting model for the simulations?

Response: The following has been added to the caption: “The initial structure for these renders is from the crystal structure of the StPSI (PDB ID: 3RFI) [4].”

Line 91: I’m not sure “motivators” is the right word here.

Response: “motivators” has been replaced with “rationale”.

Lines 95-98. I don’t think it’s necessary to justify MD as a useful technique. I would remove these lines and replace them with a few sentences on previous MD work with saposin superfamily proteins.

Response: These lines have been removed, and several references to our previous work with the StPSI, and of other SAPLIPs have been addes (L108-114).

Line 101: again so we don’t have to track back through the referenced paper – was reference 11 an experimental structure or an MD simulation?

Response: We have indicated this structure was from an experiment (L121).

Line 121-122: I’m not sure what “complementary to removal of these disulfide bonds singularly” means – I suggest rewording.

Response: We have reworded this sentence. It now reads “Whether singularly or together, both Cys6-Cys99 and Cys31-Cys71 Sγ atoms maintain a stable separation once the disulfide bond is removed” (L142-143). 

Line 137 and figure 2: refers to “shaded region above and below curves in Fig 2”. I can’t see a shaded region. Clarify.

Response: This figure has been re-plotted as a histogram with this comment taken into account when re-writing this section in light of the new figure. 

Line 169 – tell me which simulation (D_) I should be looking at (so that I don’t have to refer back to the supplementary table to find which panel in figure 3 to look at.

Response: The specific simulation has been indicated (D3). A table with this information has been added to the methods (Table 2, L518) and the supplemental table removed.

Line 187-188, “the maxima and minima are for the AI and AR simulations, respectively”. Are you sure? I can’t see any evidence for this. Can you clarify what I should be looking at.

Response: We have clairified that this is explicitly for RMSD without the loop portion (L225), and indicated the related figure (Figure 4A) (L226). Most of the values for RMSD without the loop are quite similar, and the same is true for Rg. To assist the readers in parsing the figure, a dotted line has been added to the plot, which is placed at the height of the AI result for either pH or RMSD/Rg.

Line 283 – typo – “report” should be “reports”

Response: The topological characterization section has been removed, therefore, this comment is no longer relevant.

Line 313 – I think it would be helpful to tell us more about the interactions that stabilize the association between the loop portion and the helical domains (i.e. are they hydrophobic, hydrogen bonds, involving specific pairs of residues, etc).

Response: We have added a table (Table 1, L336) to the results section following the results of Figure 6 that contains details on the intra-loop interactions, and the interactions between the loop and non-loop portions of the structure. It was found that in both cases, hydrogen bonds are the main interactions that stabilize the loop portion. A short addition to the Results section was added to detail the results of this new analysis (L320-334).

Line 317 – what is the “unperturbed state” being referred to?

Response: We wanted this to mean “without action of the StPSI on membranes or other binding partners, and without being acted on by other in vivo components (e.g., proteases, etc.). This sentence has been reworded in order to better clarify the latter.

Revised text: “Based on the above results, the disulfide bonds have no apparent structural role in a state lacking influence from other extrinsic factors (e.g., membranes, cellular components, etc.).” (L377-379).

Line 341 – did you see any differences in the simulations at the two pHs?

Response: No. Qualitatively, the results of all analyses between both pHs are quite similar. The manuscript was written with greater emphasis on the acidic pH simulations, where the StPSI is known to be active.

Line 348-349 – re-write to avoid the use of two “however”s in one sentence.

Response: One of the “however”s has been removed.

Line 373 – typo “In” should be “in”

Response: Corrected.

Line 395 and 420 – what type of “specificity” is being referred to? Specificity for a particular membrane composition? Or substrate?

Response: Membrane specificity; this has been indicated in the revision (L456, L482).

Line 432 – does this work really tell you about “function”? I think this needs to be softened – maybe just to “potential” function.

Response: The function comment has been removed. Although there is experimental evidence these two proteins have a similar function, this was not the focus of this research.

Line 457. I would have appreciated one sentence – either here or in the results – explaining your main reasons for picking this particular force field.

Response: An explanation has been added (L522-524).

General… it would be nice to have a sequence alignment (probably in the supplementary figures) that shows the alignment between StPSI and the other saposins referred to throughout the text – e.g. Saposin C, etc.”

Response: A supplementary figure (S1 Fig) has been added to better clarify a) the domain-swapped concept in regards to SAPLIPs, and b) the sequential homology between saposins and PSIs.

Figure R1: Cysteine Cα atom pair separations. The last 10% of the three 100 ns simulations were used to compute the average ± standard deviation (n = 6) for each histogram bin (blue); the last 10% of the 1 μs simulation were used to compute the average ± standard deviation (n = 2) for each histogram bin (orange) at pH 3.0 or pH 7.4. 

Figure R2: Center-of-Mass (COM) separation between helical or monomeric elements at pH 3.0 (left) and pH 7.4 (right). MA – monomer A; MB – monomer B; NL – loop residues (40-63) was excluded from the calculation; WL – loop residues were included for the calculation; AX – helix X, ranging from 1 – 4, in monomer A; BX – helix X, ranging from 1 – 4, in monomer B. The last 10% of the simulations were used to calculate average.

---

## [Decision Letter · Decision Letter 1]

5 Aug 2020

The role of disulfide bonds in a Solanum tuberosum saposin-like protein investigated using molecular dynamics

PONE-D-20-12645R1

Dear Dr. Yada,

We’re pleased to inform you that your manuscript has been judged scientifically suitable for publication and will be formally accepted for publication once it meets all outstanding technical requirements.

Kind regards,

Oscar Millet

Academic Editor

PLOS ONE

Additional Editor Comments (optional):

Reviewers' comments:

Reviewer's Responses to Questions

**Comments to the Author**

1. If the authors have adequately addressed your comments raised in a previous round of review and you feel that this manuscript is now acceptable for publication, you may indicate that here to bypass the “Comments to the Author” section, enter your conflict of interest statement in the “Confidential to Editor” section, and submit your "Accept" recommendation.

Reviewer #1: All comments have been addressed

Reviewer #2: All comments have been addressed

2. Is the manuscript technically sound, and do the data support the conclusions?

Reviewer #1: Partly

Reviewer #2: Yes

3. Has the statistical analysis been performed appropriately and rigorously? 

Reviewer #1: N/A

Reviewer #2: N/A

4. Have the authors made all data underlying the findings in their manuscript fully available?

Reviewer #1: Yes

Reviewer #2: Yes

5. Is the manuscript presented in an intelligible fashion and written in standard English?

Reviewer #1: Yes

Reviewer #2: Yes

6. Review Comments to the Author

Reviewer #1: I am pleased with the changes made by the authors. The manuscript, in this new form, can be publishable.

Reviewer #2: (No Response)

7. PLOS authors have the option to publish the peer review history of their article (what does this mean?). If published, this will include your full peer review and any attached files.

Reviewer #1: No

Reviewer #2: **Yes: **Valerie Booth

---

## [Editor Report · Acceptance letter]

12 Aug 2020

PONE-D-20-12645R1 

The role of disulfide bonds in a Solanum tuberosum saposin-like protein investigated using molecular dynamics 

Dear Dr. Yada:

I'm pleased to inform you that your manuscript has been deemed suitable for publication in PLOS ONE. Congratulations! Your manuscript is now with our production department. 

Kind regards, 

on behalf of

Dr. Oscar Millet 

Academic Editor

PLOS ONE